# Unified platform for genetic and serological detection of COVID-19 with single-molecule technology

**Noa Furth**[1][☉], **Shay Shilo**[1][☉], **Niv Cohen**[1][☉], **Nir Erez**[1], **Vadim Fedyuk**[1], **Alexander M. Schrager**[2], **Adina Weinberger**[3], **Amiel A. Dror**[4,5], **Asaf Zigron**[5,6], **Mona Shehadeh**[5,7], **Eyal Sela**[4,5], **Samer Srouji**[5,6], **Sharon Amit**[8], **Itzchak Levy**[8,9], **Eran Segal**[3], **Rony Dahan**[10], **Dan Jones**[11], **Daniel C. Douek**[2], **Efrat Shema**[1]*

1 Department of Biological Regulation, Weizmann Institute of Science, Rehovot, Israel, 2 Human Immunology Section, Vaccine Research Center, National Institutes of Health, Bethesda, MD, United States of America, 3 Department of Molecular Cell Biology, Weizmann Institute of Science, Rehovot, Israel, 4 Department of Otolaryngology, Head and Neck Surgery, Galilee Medical Center, Nahariya, Israel, 5 The Azrieli Faculty of Medicine, Bar-Ilan University, Safed, Israel, 6 Oral and Maxillofacial Department, Galilee Medical Center, Nahariya, Israel, 7 Clinical Laboratories Division, Clinical Biochemistry and Endocrinology Laboratory, Galilee Medical Center, Naharia, Israel, 8 Sheba Medical Center, Ramat Gan, Israel, 9 Sackler Medical School, Tel Aviv university, Tel Aviv, Israel, 10 Department of Immunology, Weizmann Institute of Science, Rehovot, Israel, 11 SeqLL, Woburn, MA, United States of America

☉ These authors contributed equally to this work.
* Efrat.shema@weizmann.ac.il

**Data Availability Statement:** All relevant data are within the paper and its Supporting Information files.

## Abstract

The COVID-19 pandemic raises the need for diverse diagnostic approaches to rapidly detect different stages of viral infection. The flexible and quantitative nature of single-molecule imaging technology renders it optimal for development of new diagnostic tools. Here we present a proof-of-concept for a single-molecule based, enzyme-free assay for detection of SARS-CoV-2. The unified platform we developed allows direct detection of the viral genetic material from patients' samples, as well as their immune response consisting of IgG and IgM antibodies. Thus, it establishes a platform for diagnostics of COVID-19, which could also be adjusted to diagnose additional pathogens.

## Introduction

The coronavirus disease 19 (COVID-19) is a highly infectious and pathogenic disease caused by severe acute respiratory syndrome coronavirus 2 (SARS-CoV-2) [1]. Due to its high transmutability, developing various diagnostic methods, based on independent technologies, which allow inexpensive and high-throughput detection of infection is crucial. The current gold-standard diagnostic tests of viral infections, such as real-time reverse transcription-polymerase chain reaction (RT-PCR) and enzyme-linked immunosorbent assay (ELISA), are dependent on multiple steps and involve enzymatic-based signal amplification. Recently, many new diagnostic approaches emerged to tackle the increasing need for better and diverse methods [2]. Those include reverse transcription coupled with nanopore sensing [3], isothermal

**Funding:** N.F. is supported by the Israel Cancer Research Fund. E.S. is an incumbent of the Lisa and Jeffrey Aronin Family Career Development chair. This research was supported by internal grants of the Weizmann Institute of Science, as well as Quinquin Foundation, The Benoziyo Fund for the Advancement of Science, The Sagol Institute for Longevity Research and The Willner Family Center for Vascular Biology.

**Competing interests:** Authors declare no competing interests.

amplification [4–6], CRISPR based methods [7, 8], next generation sequencing based methods [9] and improvement of RT-PCR timing through plasmonic thermocycling [10]. These novel approaches improve the time, costs and accessibility of the tests, although still mostly rely on enzymatic processes. As for serological tests, the additions of photonic ring immunoassays and bead-based digital ELISA show promising results in the ability to simultaneously measure the level of multiple antibodies against multiple antigens [11, 12].

After decades of development, single-molecule imaging techniques have matured to impact many biomedical applications, from high throughput sequencing technologies to sensitive detection of proteins [13–18]. As demonstrated by us and others [18, 19], Total Internal Reflection Fluorescence (TIRF) microscopy allows detection of single fluorophores attached to a solid surface and provides spatial and spectral multiplexing, along with quantitative detection of various molecules.

We present a proof-of concept for the use of streptavidin-biotin surface capturing, coupled with fluorescent labeling, to detect viral RNA as well as anti-viral serum antibodies by single-molecule imaging. Both approaches were evaluated on contrived and clinical samples. While development is still at need, the implementation of this technology, which is highly scalable and does not relay on enzymatic reactions, may greatly improve diagnostic capabilities of viral infections.

## Materials and methods

### Samples

RNA samples were obtained through a collaboration with Galilee Medical Center, according to hospitals' protocols. The Galilee Medical Center (GMC) implements mandatory screening swabbing related to COVID-19 for all people who come to the institution in concordance with Israel Ministry of Health requirements. This study was granted exemption from Institutional Review Boards (IRB) approval for utilizing discarded pooled RNA samples, anonymized and de-identified, for single-molecule detection of SARS-CoV-2 RNA with a multiplex approach.

RNA from nasopharyngeal samples were extracted by either eMAG instrument or QIAcube instrument under the responsibility of GMC laboratory and according to GLP. For eMAG based extraction 500µl nasopharyngeal sample were added to into 2mL lysis buffer (280134). Samples were transferred into primary tubes and analyzed according to workflow 4 (eMAG extraction methods–user manual)–"Automated transfer off-board pre-lysed samples for respiratory samples". Lysed samples were transferred automatically into vessels. 50µl silica was added for each sample. Following washes RNA was eluted in 50µl. For QIAcube based extraction 300µl nasopharyngeal sample were transferred to 2ml screw cup tubes and 400µl AVL buffer was added. After a short vortex and spin-down, samples were incubated for 20 minutes at room temp. Tubes were then transferred into a QIAcube instrument. And "Qiaamp viral RNA mini kit_manual lysis" protocol was used.

Serum samples (n = 16) of recovered COVID-19 patients were obtained from MDA (Magen David Adom, the Israeli Red Cross equivalent). These samples had been collected between March and May 2020 from non-severe cases, who had not been hospitalized. All patients were initially tested positive by RT-qPCR, and before sampling, patients had tested twice negative by RT-qPCR testing. Seropositivity of these samples had been confirmed by MDA with a commercial antibody test (Abbot, SARS-CoV-2 IgG, ref. 6R86-22/6R86-32). Research with the COVID-19 serum samples has been approved by the Weizmann Institute of Science's institutional review board (#1030–4 and #1012–1).

Use of plasma samples by NIH investigators was approved under conditions set out in an Emergency Use Simple Letter Agreement signed by The Sheba Fund for Health Services & Research and by local ethical board (protocol 7160-20-SMC). All samples were coded and de-

identified as specified in the informed consent and the NIH investigator attestation addressing the protection of human subjects and approved by the NIH Office of Human Subjects Research Protections (OHSRP). NIH research teams and the Sheba medical center teams had no access to identifiers or ability to reidentify subjects at any point.

Blood samples from healthy individuals (no prior COVID-19 infection) were collected at the Weizmann Institute of Science (IRB: 1097–2) in VACUETTE® K3 EDTA tubes and transferred immediately to ice. The blood was centrifuged (10 minutes, $1500 \times g$, 4˚C), the supernatant was transferred to fresh 50ml tubes and centrifuged again (10 minutes, $3000 \times g$, 4˚C). Supernatant was flash frozen and stored at -80˚C for long storage.

## Computational pipeline for DNA probes design and establishment of genome wide potential probe datasets

Probes for the single-molecule genetic test were selected according to the following parameters: Tm>55C, length 25–40 nt, no stable secondary structure (pairing <4 nt), and no stable capture-detection hetero-dimer. Those parameters were coded to a MATLAB script that scanned the reverse complement sequence of the SARS-CoV-2 genome (NC_045512.2) in sliding windows of 25–40 nt. Potential probes answering the criteria were collected to a dataset (S1 Table). To increase efficacy of the probes by minimizing the hybridization to human genetic sequences the probes in the dataset were scored according to the number of targets in the human genome (GRCh38) found with BLAST (blastall) alignment. Selected probes for the experimental work were also manually examined for the identity and the genomic and transcriptomic coordinates of the hits to avoid false positive result caused by hybridization of the capture and detection probes to the same molecule. The number of hits to the human genome was incorporated to a second dataset curates all possible capture and detection probe pairs with distance <25 nucleotides. The probe pair dataset attributes every probe in the pair with its coordinates, length, Tm, and BLAST hits. And every pair of probes is attributed with the number of bases that may create heterodimer, the distance between the probes, and the collective number of BLAST hits for the pair (S2 Table).

Probe sets used for this study are listed below.

| Pair | Gene | Capture/ Detection | Start (cDNA) | Stop (cDNA) | length | seq | Tm |
|---|---|---|---|---|---|---|---|
| 1 | N | capture | 955 | 995 | 40 | BiotinTGTCAAGCAGCAGCAAAGCAAGAGCAGCATCACCGCCATTG | 70.5 |
| | | detection | 1010 | 1044 | 34 | Cy5AGGAGAAGTTCCCCTACTGCTGCCTGGAGTTGAAT | 66.8 |
| | | Positive control | | | 90 | ATTCAACTCCAGGCAGCAGTAGGGGAACTTCTCCTGCTAGAATGGCTGGCAATGGCGGTGATGCTGCTCTTGCTTTGCTGCTGCTTGACA | |
| 2 | ORF1ab | capture | 10740 | 10770 | 30 | BiotinGTCGAATGTGTGGCATAAGAATAGAATAAT | 56.5 |
| | | detection | 10855 | 10890 | 35 | Cy5GCTTTAGGGTTACCAATG TCGTGAAGAACTGGGAAT | 64.4 |
| | | Positive control | | | 139 | ATTCCCAGTTCTTCACGACATTGGTAACCCTAAAGCTATTAAGTGTGTACCTCAAGCTGATGTAGAATGGAAGTTCTATGATGCACAGCCTTGTAGTGACAAAGCTTATAAAATAGAAGAATTATTCTATTCTTATGCCACACATTCTGAC | |
| 3 | ORF1a | capture | 21551 | 21587 | 36 | BiotinTACAACTATCGCCAGTAA CTTCTATGTCAGATTGATGTGA | 68.9 |
| | | detection | 21628 | 21667 | 39 | Cy5CGCACTACAGTCAATACAAGCACCAAGGTCACGGGGT | 63.5 |
| | | Positive control | | | 117 | TCACATCAATCTGACATAGAAGTTACTGGCGATAGTTGTAATAACTATATGCTCACCTATAACAAAGTTGAAAACATGACACCCCGTGACCTTGGTGCTTGTATTGACTGTAGTGCG | |
| 4 | ORF1a | capture | 26634 | 26674 | 40 | BiotinATAGTAGTTGTCTGATTGTCCTCACTGCCGTCTTGTTGACC | 67.5 |
| | | detection | 26682 | 26722 | 40 | Cy5TGTTGACTATCATCATCTAACCAATCTTCTTCTTGCTCTTC | 63.5 |
| | | Positive control | | | 89 | GAAGAGCAAGAAGAAGATTGGTTAGATGATGATAGTCAACAAACTGTTGGTCAACAAGACGGCAGTGAGGACAATCAGACAACTACTAT | |
| 5 | ORF3a | capture | 4166 | 4206 | 40 | BiotinGCAAGAAGTAGACTAAAGCATAAAGATAGAGAAAAGGGGCT | 64.5 |
| | | detection | 4215 | 4249 | 34 | AF647AGCAGCAACGAGCAAAAGGTGTGAGTAAACTGT TA | 63.3 |
| | | Positive control | | | 84 | TAACAGTTTACTCACACCTTTTGCTCGTTGCTGCTGGCCTTGAAGCCCCTTTTCTCTATCTTTATGCTTTAGTCTACTTCTTGC | |
| 6 | ORF1ab/ helicase | capture | 13310 | 13335 | 25 | BiotinAATCACCAGCATTTGTCCAGTCACAT | 56.4 |
| | | detection | 13354 | 13380 | 26 | AF647TCAGTAACATTATCGCTACCAACACAT | 55.2 |
| | | Positive control | | | 71 | ATGTGTTGGTAGCGATAATGTTACTGACTTTAATGCAATTGCAACATGTGACTGGACAAATGCTGGTGATT | |

## Genetic test sample preparation

Synthetic COVID-19 DNA in different concentrations or 10ul RNA extracted from nasopharyngeal swab samples were mixed with 1nM capture probes, 0.5nM detection probes, 0.3ul RNase inhibitor (SUPERaseIn RNase Inhibitor, AM2694, ThermoFisher), and 2X SSC buffer in a final volume of 14.8ul. Samples were incubated for 1.5h at 55C.

For synthetic COVID-19 DNA, cellular RNA extracted from HEK293 cells were added in a final concentration of 0.1ng/ul. RNA was isolated using the NucleoSpin kit (Macherey Nagel).

## RBD labeling with biotin

100ul of purified RBD (100 μM in PBS) were incubate with 1mM DTT for 15 minutes. Then, DTT was washed 3X with degassed PBS, using 3 kDa cut off Amicon concentrator (Milipore, UFC800324). Next, 250ul of reduced RBD were incubated with 10mM Melamide-Biotin and 0.1mM TCEP (ThermoFisher, 77720), overnight at 4C in the dark. On the next day, RBD-biotin was cleaned using Bio-Rad desalting column (Bio-Spin P-6 columns, 732–6002), and protein concentration was measured using NanoDrop 2000.

## Serological test sample preparation

Serum samples were diluted 1:25 in HEPES imaging buffer (12mM HEPES, 40mM TRIS pH 7.5, 60mM KCL, 0.32mM EDTA, 3mM $MgCl_2$, 10% glycerol, 0.1mg/ml BSA, 0.02% Igepal) and incubated with 100nM RBD-biotin for 1.5 hours at room temperature.

## Surface preparation for single-molecule imaging

PEG-biotin microscope slides were prepared as follows: Ibidi glass coverslips (25 mm x 75 mm, IBIDI, IBD-10812) were cleaned with (1) MilliQ H2O (3X washes, 5 minutes sonication, 3X washes), (2) 2% Alconox (Sigma 242985) (20 minutes sonication followed by 5X washes with MilliQ H2O), (3) 100% Acetone (20 minutes sonication followed by 3X washes with MilliQ H2O). To ensure surface functionality, slides were incubated in 1M KOH solution for 30 minutes while sonicated (Sigma 484016), followed by 3X washes with MilliQ H2O. Slides were sonicated for 10 minutes in 100% HPLC ethanol (J.T baker 8462–25) prior to applying amino-silanization chemistry. Slides were incubated for 24 minutes in a mixture of 3% 3-Aminopropyltriethoxysilane (ACROS Organics, 430941000) and 5% acetic acid in HPLC EtOH), with 1 minute sonication in the middle. Slides were then washed with HPLC EtOH (3X) and MilliQ H2O (3X) and dried with nitrogen. The first step of passivation was performed by applying mPEG:biotin-PEG solution (20mg Biotin-PEG (Laysan, Biotin-PEG-SVA-5000), 180mg mPEG (Laysan, MPEG-SVA-5000) dissolved in 1560μl 0.1M Sodium Bicarbonate (Sigma, S6297) on one surface followed by the assembly of another surface on top. Each pair of assembled surfaces were incubated overnight in a dark, humid environment. On the next day, surfaces were washed with MilliQ H2O and dried with N2 followed by the second passivation step. MS (PEG) 4 (ThermoFisher Scientific, TS-22341) was diluted in 0.1M of sodium bicarbonate to a final concentration of 11.7 mg/ml and applied on one surface, followed by the assembly of another surface on top. Each pair of assembled surfaces were incubated overnight in dark humid environment, washed with MilliQ H2O and dried with nitrogen. After Nitrogen flush, surfaces were stored in -20˚C.

## Single-molecule imaging by total internal reflection (TIRF) microscopy

PEG-biotin coated coverslips were assembled into Ibidi flowcell (Sticky Slide VI hydrophobic, IBIDI, IBD-80608) or 12 lanes custom made flowcells manufactured by ChipShop.

Streptavidin (SIGMA, S4762) was added to a final concentration of 0.2mg/ml followed by an incubation of 10 minutes.

For the genetic test, the PCD&PCA oxygen scavenger system was added. 3,4--Dihydroxy-benzoic acid, Protocatechuic acid (PCA) was dissolved in water to a concentration of 50mg/ml. Protocatechuate 3,4--Dioxygenase (PCD) was dissolved in 100mM Tris pH 7.5, 50mM NaCl, 50% glycerol to a concentration of 5μM. Prior to sample loading on the surface the two reagents were mixed in a 1:1 ratio and added to the sample in a 1:25 v/v ratio. Surface was washed with 2XSSC, and the reaction mixture was added to the flowcell twice, 10 minutes incubation each.

For the serological test, surfaces were blocked with 5% BSA (in PBS) for 30 minutes prior to the addition of streptavidin. Reaction mix was diluted 1:100 and loaded to flowcell twice, 5 minutes incubation each. Following additional 15 minutes incubation, the surface was washed with imaging buffer three times. Secondary anti-human IgG1 and IgM labeled antibodies (Rabbit monoclonal [H26-10] Anti-Human IgG1 H&L, Alexa Fluor® 647, Abcam, AB-ab200623 and Rabbit Anti-Human IgM mu chain (Alexa Fluor® 488), Abcam, AB-ab150189) were diluted 1:10,000 and added on the surface for 30 minutes incubation. All positions (40 FOV per experiment) were then imaged by a total internal reflection (TIRF) microscope by Nikon (Ti2 LU-N4 TIRF). Number of spots in each FOV was calculated using CellProfiler [20].

RNA clinical samples were measured in one or two separate experiments, depending on available sample volume. Serological samples were measured in two separate experiments. For each sample, quantification of all FOV from one of the experiments is shown.

## ELISA

20μl of RBD protein (2ug/ml in PBS) were added to each well of 96-well, half area high binding plate (Greiner Bio-One, cat#675061). After overnight incubation at 4˚C, the plate was washed (X3) with 0.05% Tween/PBS and incubated with 100μl blocking solution (2% FCS) for 2 hours at room temperature. The blocking solution was replaced by 20μl sample dilutions (1:100, 1:300, 1:900 and 1:2,700 in 2% FCS). After 2 hours incubation at room temperature, the plate was washed (X3, 5 minutes incubation for each wash) with 0.05%Tween/PBS. 20μl goat anti-human-HRP (Jackson 109-035-088) secondary antibody, diluted 1:2,500 in 2% FCS, was added to each well. After 1 hour incubation in room temperature, the plate was washed (X3, 5 minutes incubation for each wash) with 0.05%Tween/PBS. 20μl TMB was added to each well. After 30–60 seconds, 20μl stop solution (H2SO4 0.18M) was added, and 450nm absorbance was read.

## Statistics

Unless noted otherwise, p-values were determined using two-tailed, two-sample t-tests.

## Results

Leveraging the detection power and quantitative nature of single-molecule imaging, we developed an approach to directly detect viral RNA by TIRF microscopy (Fig 1A). Our method consists of three steps: (1) In-tube hybridization—the viral RNA is hybridized with two types of complementary DNA probes: capture probes labeled with biotin, and detection probes labeled with a fluorophore; (2) Immobilization—following hybridization, samples are added to a flow cell that contains a streptavidin-coated coverslip, allowing capture of hybridization complexes by biotin-streptavidin interaction; and (3) Imaging–the anchored complexes are imaged by TIRF microscopy, with no need of extra washing steps. Each spot in the captured image

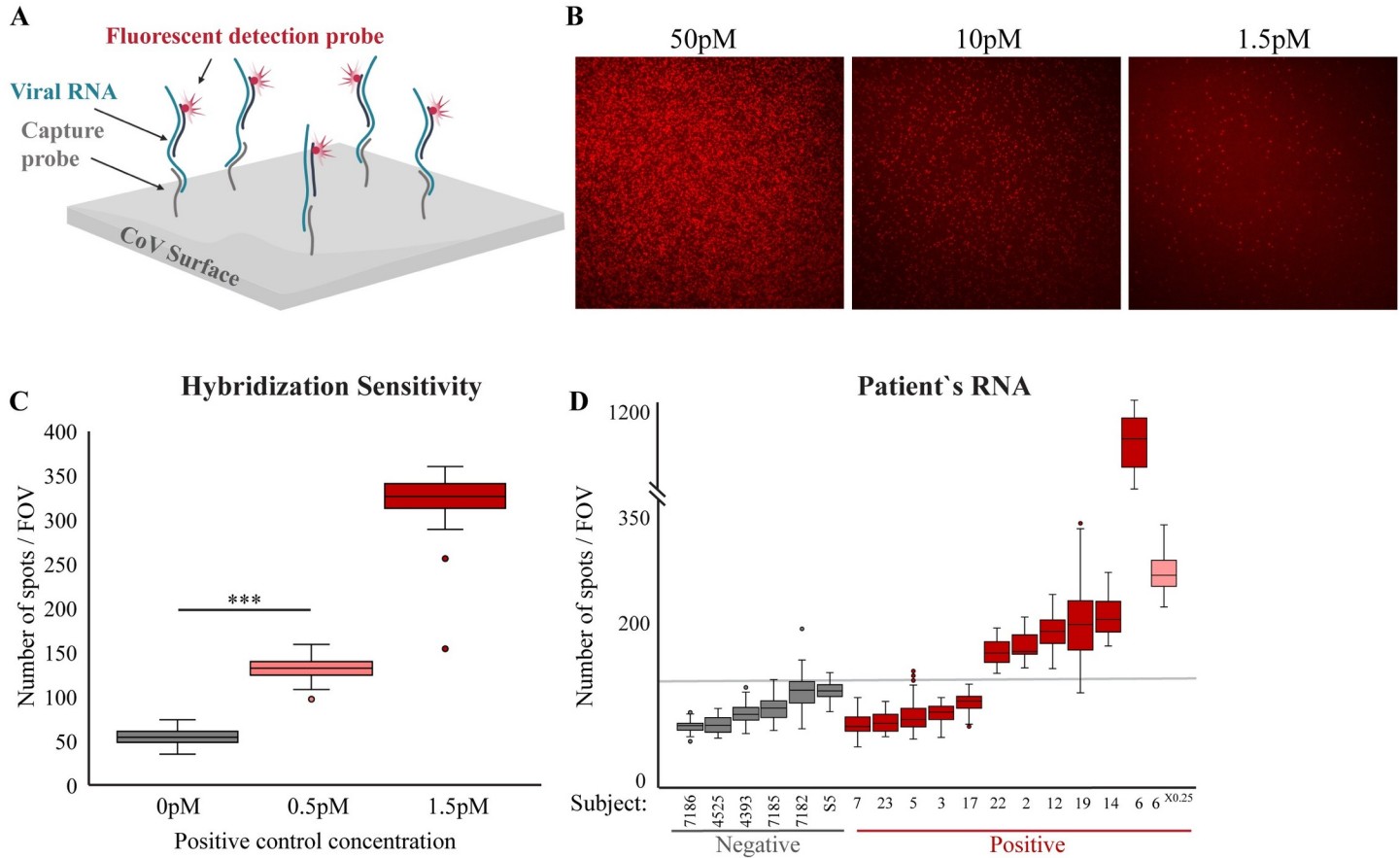

**Fig 1. Single-molecule enzyme-free detection of SARS-CoV-2 genetic material. (A)** Scheme of the genetic diagnostic test. Viral RNA is incubated with complementary DNA probes that are labeled with either biotin (capture probes) or fluorophore (detection probe). Following incubation, samples are loaded on a streptavidin-coated coverslip and imaged with TIRF microscopy. **(B)** Raw TIRF field of view (FOV) images of titration of positive control DNA at the indicated concentrations. **(C)** Quantitative and sensitive detection of hybridization complexes. Synthetic DNA controls (in the concentrations indicted in the x-axis labels) was analyzed as in B and the fluorescent signal quantified. Box plot shows the number of spots per FOV for all the FOV imaged for each sample in this experiment. For background assessment, capture and detection probes were incubated with no synthetic DNA. *** p-value <0.001. **(D)** RNA extracted from swab samples obtained from two medical facilities (samples numbers are noted on the x-axis) was analyzed as described in A. Box plot shows the number of spots per FOV for all the FOV imaged for each sample in this experiment. The highly positive sample (sample #6) was analyzed both at 1X (dark red) and 0.25X (pink) concentration. Group statistics: all negative samples: mean 91.3, Coefficient of Variation (CV) 0.2; all positive samples: mean 218.8, CV 1.3.

corresponds to a single molecule of viral RNA, as only complexes in which the viral RNA molecule stably bridged the fluorophore to the excitation region on the surface are detected. Thus, the number of spots imaged per field of view (FOV) correspond to the concentration of viral RNA in the sample.

To test the imaging sensitivity, regardless to the efficacy of the hybridization process, we imaged synthetic DNA oligomer labeled with biotin and Cy5 fluorophore. Significant signal above background values was observed at a minimal oligo concentration of 100fM (S1A Fig), with more robust and higher signal for DNA at 1pM concertation.

SARS-CoV2 specific capture and detection probes were designed according to similar principles used for microarray analysis to allow hybridization of the target molecule to two different probes. A computational pipeline was established to pick probes from the SARS-CoV2 genome sequence. The algorithm starts with creating a reverse complement (RC) sequence of the genome. Then the RC genome is scanned in sliding windows that correspond to the desired probe sizes (20–45 nucleotides). Every potential probe is examined for its Tm (>55C),

and its inability to create stable hairpin structures and self-dimers. All probes that passed this filtering are listed in S1 Table. Probes were then divided to pairs with minimal distance between binding sites (<25 nt), to offset for possible viral RNA fragmentation. Next, the sets were scored according to minimal distance between binding sites and minimal number of matches to the human genome and transcriptome (S2 Table). A total of six pairs of probes that span throughout the viral genome, were used and are listed in the Materials and Methods section. For each of the probe sets a complementary DNA oligo that match the sequence of the viral RNA in the region covered by the probes was designed, to be used as a positive control.

Series of hybridizations using titrated concentrations of the positive control DNA mixed with high concentration of human RNA showed reliable detection, even at concentration of 0.5pM (Fig 1B and 1C). This highlights the system's sensitivity to detect specific nucleic acids via hybridizations even in the presence of many off-target molecules, as would be the case in clinical samples. The decay in the signal (spots/FOV) was 2.52 fold for a 3-fold difference in concentration, illustrating the quantitative nature of the system. Of note, background signal might vary slightly between experiments due to differences in surface passivation, and thus is measured for each biotin surface used. We further verified the sensitivity and accuracy of our system by detecting commercially available synthetic whole genome CoV RNA at low concentrations (~1.5pM, S1B Fig). To test if the single-molecule system is compatible with minimal sample volume input, a single 1μl drop in the concentration of 1.5pM of CoV synthetic DNA was immobilized to the surface. We were able to successfully differentiate the sample drop from the control (S1C Fig), rendering it compatible with high-throughput microarray-based methodologies.

To assess the system's performance on clinical samples, we analyzed RNA samples from 17 nasal/oropharyngeal swabs, 6 patients with negative and 11 with positive diagnosis according to standard qPCR testing. The signal obtained from the negative samples was low and uniform (median values 70–115.5 spots/FOV). Therefore, we applied a cutoff of above a median value of 126 spots/FOV to classify samples as positive with no false-detection of negative samples as positive (Fig 1D). Within the positive samples the concertation of the viral genome greatly varied as indicated by threshold cycle (Ct) values determined for gene E by standard qPCR (S3 Table). For samples with relatively high viral RNA levels (Ct<30), the single-molecule measurements (spots/FOV) correlated reasonably with the qPCR results (Pearson correlation = -0.8). Since Ct is inversely correlated with the sample concentration, a negative correlation is expected (S3 Table). Yet, sporadic detection of samples at lower limits (Ct = 33) was also observed. Nevertheless, the sensitivity did not reach the level of qPCR, with false classification of 5 samples that were classified as positive by qPCR.

Importantly, the single-molecule data proved to be highly quantitative also when analyzing clinical samples, thus providing means for linear comparison between samples. For example, diluting a patient's RNA sample four fold resulted in a similar reduction in the number of fluorescent molecules quantified (1070±128 to 262±26 median number of spots/FOV, Fig 1D, sample #6). To summarize, the single-molecule methodology is a simple, non-enzymatic based, route to directly measure viral RNA in samples with high viral load.

To expand our single-molecule technology for COVID-19 diagnostics, we leveraged its inherent capacity for simultaneous detection of various types of molecules that are spatially separated on the surface. Specifically, we aimed to supplement viral RNA detection with the detection of antibodies in patients' serum. Serological tests have been shown to complement genetic tests, since antibodies accumulate several days post symptom onset [21], when the efficacy of genetic tests drops [22, 23]. These tests are also critical for evaluating the potential spread of the disease, and guiding public policies related to the pandemic. An ideal test will (a) target the receptor-binding domain (RBD) of the spike protein that is more likely to be

indicative for the presence of neutralizing antibodies, and can further serve to monitor vaccine effectiveness [24], and (b) be able to differentiate between ongoing infection (IgM antibodies) and late/post-infection immunity (IgG antibodies).

To quantify viral-specific antibodies present in the serum, some adaptations were made in the single-molecule approach, although the general principles are similar to those described for the genetic test. Tagging the spike protein RBD domain (AA 319–514, [21]) with biotin and incubating it with serum/plasma allows us to capture circulating antibodies on the microscope coverslip. Following binding of complexes, the surface is washed to remove the unbound molecules. Next, fluorescently labeled anti-human IgG/IgM antibodies are used to detect the captured antibodies (Fig 2A). TIRF microscopy is used to image the surface, and every detected spot corresponds to a single antibody complex.

Titration of recombinant anti-RBD IgG antibodies show that the system can quantitatively detect varying amounts of antibodies, reaching a Limit of Detection (LoD) of 0.5 pM concentrations (Fig 2B). Quantification of the signal decay fitted to the concentration differences ($R^2$ = 0.9, logarithmic fit).

We next probed a panel of serum samples collected from convalescent patients (n = 16, who had a previous positive COVID-19 PCR test, see Materials and Methods section) or healthy subjects without a known prior COVID-19 infection (n = 9) as negative samples. The negative samples showed low and uniform signal (median values of 43.5–86 spots/FOV), determining a threshold (median of 90 spots/FOV) to differentiate negative from positive samples. Anti-RBD IgG antibodies were positively detected in the serum of 15 out of the 16 convalescent patients, pointing toward sensitivity of 93.8% (Fig 2C), which is in line with high quality approved tests. Interestingly, we detected extremely high variability in the number of IgG antibodies in the sera of the convalescent patients, with a dynamic range reaching up to 95 fold between individuals with low versus high IgG levels. This is particularly interesting as all convalescent samples were collected from individuals who had mild COVID-19 symptoms that did not require hospitalization.

To further evaluate the single-molecule serological test performance, we probed all serum samples with a classical ELISA assay against RBD antibodies (Figs 2D and S2). The results of the two test correlated reasonably (Pearson correlation = 0.62). Interestingly, while the single-molecule system and ELISA showed similar trends across different samples, the dynamic range observed for the single-molecule assay is significantly higher; while ELISA showed up to 4.5 folds change from control samples, some of the samples reached up to 95 folds when measured by single-molecule. Furthermore, comparison with ELISA results confirmed the high sensitivity of the single-molecule assay. The convalescent sample that scored negatively by single-molecule measurements (sample 8) also showed negative values with ELISA; we therefore suspect that this sample lacked anti-RBD antibodies. The lack of anti-RBD antibodies in 2% of patients 30 days after symptoms onset has been described [25]. Moreover, the ELISA test failed in detecting four additional samples with low antibody levels that were correctly classified by the single-molecule test (Fig 2C and 2D, samples 1, 13, 15, and 16).

The single-molecule imaging approach allows multiplexing the detection of different antibody isotypes in the same samples by taking advantage of spectral separation of fluorophores. To explore this possibility we examined serum samples from five patients with an active disease, four in the range of 4–9 days since symptoms onset and one asymptomatic patient at the time of testing (S4 Table). The probing was done simultaneously for human IgM antibodies labeled with Alexa Fluor 488 (Fig 2E) and human IgG labeled with Alexa Fluor 647 (Fig 2F). All the samples from symptomatic patients were found to have high levels of IgM antibodies even as short as 4 days after symptoms onset. Yet, the system failed to detect the a-symptomatic patient (#25), which was tested only 2 days after qPCR detection of COVID-19.

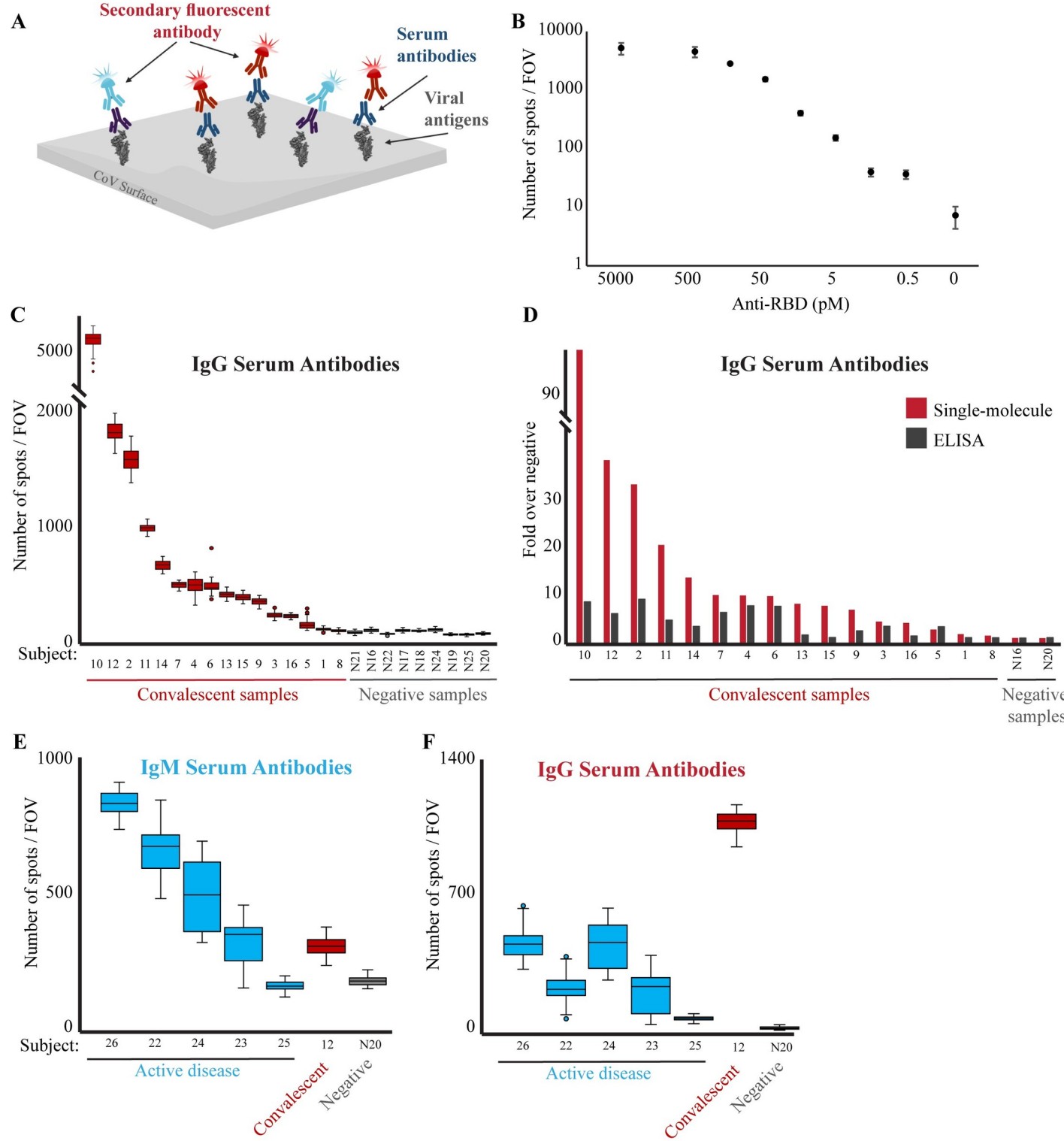

**Fig 2. Single-molecule detection of anti-RBD antibodies.** (**A**) Scheme of the serological diagnostic test. Serum samples are incubated with biotin-conjugated viral antigen (RBD) and loaded on a PEG-coated, streptavidin activated coverslip. Multiplex of fluorescently-labeled anti-human IgG (red) and IgM (light blue) antibodies are added to the flow cell and imaged. (**B**) Human anti-RBD antibodies at the indicated concentrations were incubated with biotin-RBD, and detected by fluorescently-labeled anti-human IgG antibodies. The Antibodies LoD is at picomolar concentrations. Both axes are in logarithmic scale, and the no anti-RBD antibody data point is not to scale. (**C**) Serum samples from either convalescent or not-infected subjects were diluted 1:2500 and analyzed as described in B to detect the presence of anti-RBD IgG antibodies in the subjects' serum. The box plot shows the number of spots per FOV for all the FOV imaged for each sample in this

experiment. Group statistics: all negative samples: mean 63.1, CV 0.3; all positive samples: mean 827.3, CV 1.5. Median values of each group were compared by t-test, p-value < 0.05. **(D)** Comparison between single-molecule and ELISA detection of anti-RBD antibodies. Single-molecule imaging and ELISA against anti-RBD antibodies were conducted on the same samples. Signals from each assay were normalized compared to the negative serum samples. Single-molecule imaging provides greater sensitivity and dynamic range in detecting anti-RBD antibodies in serum. **(E, F)** Serum from subjects with an active COVID-19 disease (blue), convalescent (red), or not-infected (gray) subjects, were diluted 1:2500, incubated with biotin-RBD and loaded on a streptavidin-coated surface. Fluorescently labeled anti-human IgM (E) or IgG (F) antibodies were imaged and quantified.

Surprisingly, despite the short period of time since symptom onset, all the samples demonstrated distinct level of IgG antibodies compared to the negative sample, although lower than the IgM levels, probably due to the early stages of infection. Indeed probing a convalescent sample with both antibodies revealed opposite trend, with high levels of IgG and low levels of IgM antibodies, confirming the specificity of the detection antibody. To summarize, we provide a proof-of-concept for single-molecule detection of different antibody isotypes in one, sensitive and simple assay.

## Discussion

Here we adapted the capabilities of single-molecule imaging technology to generate a unified platform capable of detecting either the pathogen's genetic material or antibodies in patients' serum. Each of these work-flows holds great potential for multiplexing; detection of several different molecules within one test, and is compatible with very low sample volumes. Importantly, our method does not require enzymatic reactions and signal amplification.

While the detection of antibodies outperforms the classical RBD-IgG ELISA assay by the measured parameters (dynamic range, sensitivity, and throughput), the sensitivity of single-molecule hybridizations falls short of amplification-based PCR reactions, which can reach a detection limit of 100–1000 copies of viral RNA per milliliter of transport media [26]. Higher sensitivity may be achieved by implementing single-molecule kinetic fingerprinting [15, 27], to be explored in future studies. Furthermore, additional studies on larger cohorts, as well as systematical comparisons to additional available serological tests are needed. Finally, combined single-molecule based detection of both viral RNA and antibodies was recently demonstrated by Ter-Ovanesyan and colleagues for saliva samples [28]. The use of saliva samples which contain both viral RNA and antibodies against viral particles holds great potential and can be further explored for analysis by our adaptable single-molecule platform.

The computational pipeline for the probe design provides a database for SARS-CoV-2 probes and is straightforward to adjust for detection of additional pathogens (see Materials and Methods). We explored the use of multiple probes targeting the same viral genome, in order to increase sensitivity. However, this approach may compromise the ability to determine the exact number of viral genomes present in a sample. A primary advantage of the single-molecule genetic test lies in its straightforward adaptability for multiplexed detection of several variants of the same pathogen, or several different pathogens; the only adjustment needed for rapid response to suspected future outbreaks, or the appearance of new variants, is the design of new probes.

Scaling-up the system to high-throughput can be implemented by using immobilization of low-volume samples in an array configuration, as demonstrated in S1C Fig [29]. Following immobilization of each sample to a predesigned specific location, all downstream steps, including hybridization, washes, and imaging, are applied to all samples simultaneously. For multiple detections of three-to-four pathogens from the same sample, spectral separation can be utilized. This is achieved by designing detection probes with a specific fluorophore for each pathogen, using a similar approach as the simultaneous detection of both IgG and IgM antibodies described above (Fig 2E and 2F). Thus, a patient diagnosed with lung infection can be tested for

the most common and/or dangerous viruses such as the novel CoV, SARS, MERS and swine influenza. Smart pooling of detection-capture probes can extend the four colors to almost any number of tests simultaneously [30, 31]. Overall, this work serves as proof-of-principal for such applications, which are expected to be relevant and instrumental for diverse clinical utilities.

## Supporting information

**S1 Fig. (A)** Single-molecule detection of Cy5-DNA probes. Biotin and Cy5 labeled DNA probes at the indicated concentrations were added to a streptavidin-coated surface and imaged by TIRF. *** p-value <0.001. **(B)** SARS-CoV-2 synthetic RNA (Twist Bioscience) was incubated with capture and detection probes and analyzed as in Fig 1B. *** p-value <0.001. **(C)** COVID-19 synthetic DNA was incubated with capture and detection probes. A 1µl drop of the hybridized sample was immobilized on a streptavidin-coated surface and imaged. For background assessment, capture and detection probes were incubated with no synthetic DNA. *** p-value <0.001.
(TIF)

**S2 Fig. Serum samples from convalescent or not-infected subjects (N16 and N20) were diluted 1:100, 1:300, 1:900 and 1:2,700 and antibodies against RBD were probes by ELISA.**
(TIF)

**S1 Table. All possible single probes derived by the computational pipeline.** Coordinates corresponding to the reverse complement strand of the COVID-19 genome, probe length, sequence and Tm are indicated.
(CSV)

**S2 Table. All possible capture and detection pairs derived by the computational pipeline.** Table includes the following information: 'Line 1': serial number for the capture probe; 'start1': starting coordinate of capture probe on reverse compliment strand of the COVID-19 genome; 'stop1': end coordinate of capture probe on reverse compliment strand of the COVID-19 genome; 'len1': length of capture probe; 'seq1': sequence of the capture probe; 'Tm1': the Tm of the capture probe; 'Line 2': serial number of the detection probe; 'start2': starting coordinate of detection probe on reverse compliment strand of the COVID-19 genome; 'stop2': end coordinate of detection probe on reverse compliment strand of the COVID-19 genome; 'len2': length of detection probe; 'seq2': sequence of the detection probe; 'Tm2': the Tm of the detection probe; 'Dimer 1+2 (bases)': number of bases that may create a stable heterodimer between the capture and detection probes; 'Blast1': number of blast hits for the capture probe; 'Blast2': number of blast hits for the detection probe; 'Blast 1+2': the sum of blast hits of capture and detection probes; 'dist': distance between the capture and detection probes (with upper limit of 25 nt).
(CSV)

**S3 Table. Ct values (gene E) and single-molecule scores (median of number of spots/FOV) for positive swab samples presented in Fig 1D.**
(DOCX)

**S4 Table. Clinical features of serum samples tested from patients with active disease.**
(DOCX)

## Acknowledgments

We thank Irit Sagi, Ori Avinoam, Noam Stern-Ginossar and Roi Avraham for fruitful discussions and advice.

## Author Contributions

**Conceptualization:** Noa Furth, Shay Shilo, Dan Jones, Daniel C. Douek, Efrat Shema.

**Formal analysis:** Noa Furth, Shay Shilo, Niv Cohen.

**Investigation:** Noa Furth, Niv Cohen, Nir Erez, Vadim Fedyuk, Efrat Shema.

**Methodology:** Nir Erez, Vadim Fedyuk.

**Resources:** Alexander M. Schrager, Adina Weinberger, Amiel A. Dror, Asaf Zigron, Mona Shehadeh, Eyal Sela, Samer Srouji, Sharon Amit, Itzchak Levy, Eran Segal, Rony Dahan, Daniel C. Douek.

**Writing – original draft:** Noa Furth, Efrat Shema.

**Writing – review & editing:** Noa Furth, Shay Shilo, Niv Cohen, Amiel A. Dror, Dan Jones, Efrat Shema.

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
