## [Decision Letter · Decision Letter 0]

22 Jun 2021

PONE-D-21-17639

Multiplexed Detection of COVID-19 with Single-Molecule Technology

PLOS ONE

Dear Dr. Shema,

Thank you for submitting your manuscript to PLOS ONE. After careful consideration, we feel that it has merit but does not fully meet PLOS ONE’s publication criteria as it currently stands. Therefore, we invite you to submit a revised version of the manuscript that addresses the points raised during the review process.

We look forward to receiving your revised manuscript.

Kind regards,

Ruslan Kalendar

Academic Editor

PLOS ONE

Journal Requirements:

" N.F. is supported by the Israel Cancer Research Fund.

E.S. is an incumbent of the Lisa and Jeffrey Aronin Family Career Development chair.

This research was supported by internal grants of the Weizmann Institute of Science, as

well as Quinquin Foundation, The Benoziyo Fund for the Advancement of Science, The

Sagol Institute for Longevity Research and The Willner Family Center for Vascular

Biology."

"N.F. is supported by the Israel Cancer Research Fund. E.S. is an incumbent of the Lisa and Jeffrey Aronin Family Career Development chair. This research was supported by internal grants of the Weizmann Institute of Science, as well as Quinquin Foundation, The Benoziyo Fund for the Advancement of Science, The Sagol Institute for Longevity Research and The Willner Family Center for Vascular Biology."

Reviewers' comments:

Reviewer's Responses to Questions

**Comments to the Author**

1. Is the manuscript technically sound, and do the data support the conclusions?

Reviewer #1: Yes

Reviewer #2: Yes

Reviewer #3: Yes

2. Has the statistical analysis been performed appropriately and rigorously? 

Reviewer #1: Yes

Reviewer #2: Yes

Reviewer #3: I Don't Know

3. Have the authors made all data underlying the findings in their manuscript fully available?

Reviewer #1: Yes

Reviewer #2: Yes

Reviewer #3: Yes

4. Is the manuscript presented in an intelligible fashion and written in standard English?

Reviewer #1: Yes

Reviewer #2: Yes

Reviewer #3: Yes

5. Review Comments to the Author

Reviewer #1: 

I have a minor comment and some questions:

1. Table S2 file is same as Table S1.

2. There is a difference in background in Figure S1B and S1C. Template type is the only difference, and background is higher when Twist RNA is the template. What could be the reason?

3. Have authors exploited the possibility of single RNA genome binding to other capture probes targeting different genome regions and how this affects quantitative feature of the assay?

Reviewer #2: 

The authors describe a method for detecting both RNA from the SARS-CoV-2 virus and antibodies to the virus in clinical samples using a single molecule counting method. The manuscript is well written and the experiments well designed and explained. The results from testing of clinical samples indicate that the tests are specific by comparisons to conventional methods such as PCR and ELISA. The research warrants publication after a few points have been addressed:

1) My main concern is that the claim in the title of the paper that the method is “Multiplexed Detection” is misleading because the two types of molecules are not measured at the same time using the same methods. While the authors have described methods for both RNA and antibodies, it is clear from the methods section that the two methods cannot be performed simultaneously because they require different sample prep methods and different buffer systems for incubation with the capture surfaces. The authors should make this clear in the Results or Discussion section. I would also recommend changing the title.

2) The authors are clear that the sensitivity of their method “falls short of amplification-based PCR reactions” but do not provide any quantitative comparisons between the methods. The authors should provide a limit of detection (LOD)—which seems to be around 0.5 pM—and compare that to PCR methods. The sensitivity of these methods are widely available online (https://www.fda.gov/medical-devices/coronavirus-covid-19-and-medical-devices/sars-cov-2-reference-panel-comparative-data).

3) Similarly the authors should indicate the improvement in sensitivity of the antibody tests compared to the ELISA that they performed.

4) The authors should cite and compare their work to a recent report of using single molecule label detection for measuring of RNA and antibodies from COVID-19 patients from Walt and co-workers https://pubs.acs.org/doi/10.1021/acs.analchem.1c00515

Reviewer #3: 

The manuscript by Efrat Shema et al. present the use of TIRF to detect SARS-Cov2 RNA and IgG and IgM antibodies derived from the humane immune response to the virus. The approach is elegant and overcome some of the hurdles that current RT-qPCR assays present, for instance, the need of extracting RNA from their biological matrixes.

While the work is scientifically sound, well written and with great potential I'd like to do the following comments:

1) Using multiplexed in the title can be confusing as the classic use of the word means the simultaneous detection of more than one analyte using the same sample. In this manuscript, the detection platform is the same for both RNA and antibodies but two different set of samples need to be used.

3) Overall, I miss the number of replicates use per sample and the CV% of each experiment.

2) To calculate the limit of detection, more than 3 points in the calibration curve (Fig. 1C) would be needed. Alternatively, it could be presented as a system with a single cut-off point for yes/no answer but not for quantifying RNA molecules.

3) While in different figures appear multiple signals for the same condition, ie, Fig 1C, it is not clear if they are different spot counts from different FOVs or from different sample replicates, please add it to Fig. legends.

4) Details of how nasopharyngeal clinical samples used to detect RNA were treated are missing. What did the authors use to break up viral particles?

5) Authors stablished the LoD at 125 spots/FOV. In Table S3 there are 5 out 11 of clinical samples with numbers below 125, hence I'd flag those samples mentioning that they are below the cut-off point.

I'd like to recommend the manuscript to be published once authors address these points.

Best regards

6. PLOS authors have the option to publish the peer review history of their article (what does this mean?). If published, this will include your full peer review and any attached files.

Reviewer #1: No

Reviewer #2: No

Reviewer #3: **Yes: **Juan J. Diaz Mochon

---

## [Author Response · Author response to Decision Letter 0]

7 Jul 2021

We are thankful for the important comments received from the reviewers. We did our best to revise the manuscript according to the reviewers’ suggestions, as described in details bellow. We hope that in light of the changes introduced, you`ll find our revised manuscript suitable for publication in PLOS ONE. 

Point-by-point response to reviewer`s requests:

Reviewer #1: 

I have a minor comment and some questions:

1. Table S2 file is same as Table S1.

We apologize for uploading the wrong file by mistake. We have corrected this, and table S2 is now available. 

2. There is a difference in background in Figure S1B and S1C. Template type is the only difference, and background is higher when Twist RNA is the template. What could be the reason?

We thank the reviewer for pointing this out and apologize for not directly addressing this in the original manuscript. The background signal varies slightly between experiments, with most of the experiments showing very low background (for example, see also figure 1C). These differences may arise from minor differences in surface passivation. For this reason, we find it critical to include a background measurement in each experiment and compare sample measurement to the relevant control. When analyzing multiple samples which required more than one surface, we made sure to use surfaces from the same preparation batch and measured both positive and negative samples on each surface to account for this important factor. We have revised the text to include these important clarifications. 

3. Have authors exploited the possibility of single RNA genome binding to other capture probes targeting different genome regions and how this affects quantitative feature of the assay?

We designed the system to include capture-detection probes targeting several different regions of the viral genome, in order to increase the sensitivity of the assay (we assume that the viral genome may be fragmented, and aim to exploit this by using multiple probes). It is likely that increasing the number of probes even further, to cover the entire viral genome, would further increase the sensitivity of the assay; this can be explored in future work. Regarding the quantitative nature of the assay: our results indicate that the signal obtained in our system correlates with viral load (Figure 1D). However, we are not able to determine the exact number of viral genomes present in a sample. We now refer to this limitation in the discussion. 

Reviewer #2: 

The authors describe a method for detecting both RNA from the SARS-CoV-2 virus and antibodies to the virus in clinical samples using a single molecule counting method. The manuscript is well written and the experiments well designed and explained. The results from testing of clinical samples indicate that the tests are specific by comparisons to conventional methods such as PCR and ELISA. The research warrants publication after a few points have been addressed:

1) My main concern is that the claim in the title of the paper that the method is “Multiplexed Detection” is misleading because the two types of molecules are not measured at the same time using the same methods. While the authors have described methods for both RNA and antibodies, it is clear from the methods section that the two methods cannot be performed simultaneously because they require different sample prep methods and different buffer systems for incubation with the capture surfaces. The authors should make this clear in the Results or Discussion section. I would also recommend changing the title.

We thank the reviewer for this important notion. According to the reviewer’s suggestion, we have revised the title of the paper to better reflect the work carried out (“Unified platform for genetic and serological detection of COVID-19 with single-molecule technology”). We also revised the abstract and discussion to clarify this.

2) The authors are clear that the sensitivity of their method “falls short of amplification-based PCR reactions” but do not provide any quantitative comparisons between the methods. The authors should provide a limit of detection (LOD)—which seems to be around 0.5 pM—and compare that to PCR methods. The sensitivity of these methods are widely available online (https://www.fda.gov/medical-devices/coronavirus-covid-19-and-medical-devices/sars-cov-2-reference-panel-comparative-data).

Thank you for this comment. We agree that it is very important to compare the sensitivity of our method to that of PCR. Thus, we have analyzed the correlation between our single-molecule data (spots per field of view) to qPCR done on the same samples (Pearson correlation =-0.8). Of note, this correlation holds only for samples with relatively high levels of viral RNA (Ct<30).

Regarding the limit of detection: our results suggest that samples in ~1pM concentration show signal which is significantly above background in our assay; indeed this is not as sensitive as amplification based assays which reach a detection limit of 100-1000 copies of viral RNA per milliliter of transport media (0.2-2aM). We have added this information to the discussion of the revised paper. 

3) Similarly the authors should indicate the improvement in sensitivity of the antibody tests compared to the ELISA that they performed.

The sensitivity of our serological assay was assessed in two ways. The first one aimed to determine the minimal concentration of anti-RBD antibodies that can be detected above background signal (Figure 2B). The second analyzed the same plasma samples (from convalescent subjects) by both the single-molecule assay and classical ELISA targeting Spike-RBD antibodies. Four samples that were detected as positive in the single-molecule assay (as well as by qPCR to validate infection) scored negative with ELISA. Furthermore, while the single-molecule assay correlated well with the ELISA results, it showed much higher dynamic range, reaching a 5-10 times higher signal in some of the samples. Within the scope of this study we compared our assay to one commonly used ELISA assay, although multiple tests are currently available ranging in their sensitivity and accuracy. In the tests performed as part of this study, the single molecule assay shows good sensitivity (~94%) which places it in line with FDA approved tests. We are well aware that it is hard to estimate the precise sensitivity of ELISA and the improvement of our assay over other available tests. This will require a larger cohort of samples and systematical comparison to multiple available serological tests. Those clarifications were added to the text. Nevertheless, in general, single-molecule methods are expected to have higher sensitivity than ELISA and hold great potential in various areas of diagnostics (reviewed recently by Nils Walter and colleagues, Acc. Chem. Res. 2021).

4) The authors should cite and compare their work to a recent report of using single molecule label detection for measuring of RNA and antibodies from COVID-19 patients from Walt and co-workers https://pubs.acs.org/doi/10.1021/acs.analchem.1c00515

We thank the reviewer for pointing us to the very important work done by the Walt group. Indeed there are many similarities between the work of Walt and colleges and ours. Both methods leverage the power of a single molecule platform for detection of viral genetic material and antibodies. Although the source of samples was different, the results are similar and indicate higher sensitivity in serological tests. We have revised the text to include this approach both in the introduction and the discussion. The use of saliva holds great promise since it includes both types of molecules in one bio-fluid. We hope to explore the use of this bio-fluid on our single-molecule platform in the future. 

Reviewer #3: 

The manuscript by Efrat Shema et al. present the use of TIRF to detect SARS-Cov2 RNA and IgG and IgM antibodies derived from the humane immune response to the virus. The approach is elegant and overcome some of the hurdles that current RT-qPCR assays present, for instance, the need of extracting RNA from their biological matrixes.

While the work is scientifically sound, well written and with great potential I'd like to do the following comments:

1) Using multiplexed in the title can be confusing as the classic use of the word means the simultaneous detection of more than one analyte using the same sample. In this manuscript, the detection platform is the same for both RNA and antibodies but two different set of samples need to be used.

We thank the reviewer for this important comment. According to the reviewer’s suggestion, we have revised the title of the paper (“Unified platform for genetic and serological detection of COVID-19 with single-molecule technology”). We also revised the abstract and text to clarify that each type of bio-molecule is analyzed separately, as the reviewer correctly points out. Importantly, within each workflow, different sequence of RNA or types of antibodies can be multiplexed. Furthermore, since the system does not require enzymatic reactions, it can be adapted to test bio-fluids that contain both types of molecules in a relatively straightforward manner.

3) Overall, I miss the number of replicates use per sample and the CV% of each experiment.

We apologize for not including this information in the original manuscript. We tested the plasma samples in two independent experiments. RNA samples were tested once or twice, depending on the sample volume available. In both cases quantification is shown for all fields of view from one of these measurements. We have added this information to the text in the methods section. The analysis of sample #6, shown in figure 1D (1X and 0.25X dilution), was conducted in two separate experiments, supporting consistency between measurements. We have also calculated the Coefficient of Variation for each group of samples (positive and negative) and added this information in the figure legends.

 2) To calculate the limit of detection, more than 3 points in the calibration curve (Fig. 1C) would be needed. Alternatively, it could be presented as a system with a single cut-off point for yes/no answer but not for quantifying RNA molecules.

The reviewer is correct. We used the synthetic DNA shown in figure 1C as a general demonstration of the sensitivity of our system (~1pM). In addition, for clinical samples we used the negative RNA samples to set an appropriate threshold for detection (median value of 126 spots/FOV, figure 1D). We believe that further improvements of the system, as well as additional analysis of multiple samples, will allow refining of this threshold. We also revised the discussion to clarify that the system’s ability to directly count complete genomes is yet to be determined.

3) While in different figures appear multiple signals for the same condition, ie, Fig 1C, it is not clear if they are different spot counts from different FOVs or from different sample replicates, please add it to Fig. legends.

Thank you for this comment. We have modified the figure legend to include this information. 

4) Details of how nasopharyngeal clinical samples used to detect RNA were treated are missing. What did the authors use to break up viral particles?

Thank you for this important comment. We have added this information to the methods section.

5) Authors stablished the LoD at 125 spots/FOV. In Table S3 there are 5 out 11 of clinical samples with numbers below 125, hence I'd flag those samples mentioning that they are below the cut-off point.

We have added this information to table S3.

---

## [Editor Report · Decision Letter 1]

12 Jul 2021

Unified platform for genetic and serological detection of COVID-19 with single-molecule technolog

PONE-D-21-17639R1

Dear Dr. Shema,

We’re pleased to inform you that your manuscript has been judged scientifically suitable for publication and will be formally accepted for publication once it meets all outstanding technical requirements.

Kind regards,

Ruslan Kalendar

Academic Editor

PLOS ONE

---

## [Editor Report · Acceptance letter]

16 Jul 2021

PONE-D-21-17639R1 

Unified platform for genetic and serological detection of COVID-19 with single-molecule technology 

Dear Dr. Shema:

I'm pleased to inform you that your manuscript has been deemed suitable for publication in PLOS ONE. Congratulations! Your manuscript is now with our production department. 

Kind regards, 

on behalf of

Prof. Ruslan Kalendar 

Academic Editor

PLOS ONE